# Multifaceted Heparin: Diverse Applications beyond Anticoagulant Therapy

**DOI:** 10.3390/ph17101362

**Published:** 2024-10-12

**Authors:** Razia Sultana, Masamichi Kamihira

**Affiliations:** 1Department of Chemical Engineering, Faculty of Engineering, Kyushu University, 744 Motooka, Nishi-ku, Fukuoka 819-0395, Japan; razia@nstu.edu.bd; 2Department of Biotechnology and Genetic Engineering, Faculty of Science, Noakhali Science and Technology University, Noakhali 3814, Bangladesh

**Keywords:** heparin, glycosaminoglycan, anticoagulant, bioengineering, nanotechnology, nano-drug delivery systems

## Abstract

Heparin, a naturally occurring polysaccharide, has fascinated researchers and clinicians for nearly a century due to its versatile biological properties and has been used for various therapeutic purposes. Discovered in the early 20th century, heparin has been a key therapeutic anticoagulant ever since, and its use is now implemented as a life-saving pharmacological intervention in the management of thrombotic disorders and beyond. In addition to its known anticoagulant properties, heparin has been found to exhibit anti-inflammatory, antiviral, and anti-tumorigenic activities, which may lead to its widespread use in the future as an essential drug against infectious diseases such as COVID-19 and in various medical treatments. Furthermore, recent advancements in nanotechnology, including nano-drug delivery systems and nanomaterials, have significantly enhanced the intrinsic biofunctionalities of heparin. These breakthroughs have paved the way for innovative applications in medicine and therapy, expanding the potential of heparin research. Therefore, this review aims to provide a creation profile of heparin, space for its utilities in therapeutic complications, and future characteristics such as bioengineering and nanotechnology. It also discusses the challenges and opportunities in realizing the full potential of heparin to improve patient outcomes and elevate therapeutic interventions.

## 1. Introduction

Heparin, a highly sulfated therapeutic agent with a long history, has been acknowledged for its remarkable anticoagulant properties [1,2,3,4,5]. It was first discovered in extracts from animal tissues in the early 20th century. Jay McLean identified its anticoagulant effect in the canine liver in 1916 [6]. It was later named “heparin” by Howell and Holt in 1922 and became pivotal in medical practice by the 1930s following advancements in purification techniques [7,8,9]. Its interaction with antithrombin was elucidated in the 1950s, leading to enhanced therapeutic understanding [10,11,12]. A brief history of heparin is presented in Figure 1 [13,14]. Since its discovery, this glycosaminoglycan (GAG) has played a pivotal role in preventing and treating thrombotic disorders, saving countless lives globally [6,15]. However, beyond its established role in anticoagulation, emerging research has unveiled a plethora of novel therapeutic pathways, including anti-inflammatory, antiviral/bacterial, antithrombotic, anti-metastatic, anti-hypolipidemic, and anti-angiogenesis activities [2,3,5,16,17,18,19,20,21].

Currently, the COVID-19 pandemic remains a critical global health challenge [22]. Ongoing research, including hypotheses, clinical data, and retrospective cohort studies, suggests that unfractionated heparin (UFH) and low-molecular-weight heparin (LMWH) may reduce mortality in COVID-19 patients by leveraging their anticoagulant, antiviral, and anti-inflammatory properties [22,23,24,25,26,27,28]. Consequently, the versatile nature of heparin has prompted a thorough exploration of its diverse applications across various medical fields.

The development of LMWH was driven by the need to create heparin derivatives that could effectively inhibit factor Xa without the extensive anticoagulant effects associated with thrombin inhibition [29,30,31,32]. Research indicates that factor Xa inhibition by heparin relies predominantly on a specific pentasaccharide sequence, while effective thrombin inhibition involves additional interactions with a longer polysaccharide chain [33,34]. Understanding these distinct mechanisms was crucial in the development of LMWHs. LMWHs are derived from heparin through depolymerization, resulting in shorter chains that preferentially inhibit factor Xa over thrombin. This structural modification retains the antithrombotic efficacy of heparin while reducing the risk of bleeding complications, making LMWHs a vital therapeutic option in various clinical settings for preventing and treating thromboembolic disorders.

Recent years have witnessed a profound expansion in the therapeutic potential of heparin, driven by advancements in comprehending its complex structure–function relationships and interactions with biological molecules [3,35,36,37,38,39]. The traditional extraction methods from animal origins have yielded heparin with a complex structure and mechanism of action that have captivated researchers for decades [40,41,42,43,44,45,46]. Its sulfated polysaccharide backbone imparts a unique molecular architecture, facilitating interactions with a range of biological molecules, from growth factors to cell surface receptors. While its anticoagulant effects remain its hallmark feature, the elucidation of the complex pharmacology of heparin has revealed a wealth of therapeutic opportunities across numerous therapeutic areas [2,4,5,47,48,49]. Moreover, the integration of heparin with nanomaterials has initiated a new era of drug delivery and functionalization, offering innovative pathways to enhance its therapeutic efficacy and bioavailability [49,50,51,52,53,54].

This review aims to provide a comprehensive overview of the expanding therapeutic potential of heparin beyond its traditional role. It underscores the pivotal role of heparin in contemporary therapeutic strategies, bolstered by cutting-edge advancements in bioengineering and nanotechnology. By exploring recent advances in understanding its biosynthesis process and clinical applications, we seek to illuminate the diverse opportunities and challenges in harnessing the full therapeutic repertoire of this remarkable molecule. From inflammation modulation to antiviral therapy, from cancer treatment to tissue regeneration, heparin stands at the forefront of modern medicine, offering numerous possibilities to improve patient outcomes and redefine therapeutic interventions.

## 2. Heparin within Glycosaminoglycans

Heparin, renowned for its critical role in clinical practice as a powerful anticoagulant, exhibits a complex molecular structure characteristic of GAGs. As a part of the GAG family, heparin consists of repeating disaccharide units composed of uronic acid [L-iduronic acid (IdoA) or D-glucuronic acid (GlcA)] and D-glucosamine (GlcN), with extensive sulfation at multiple positions. Heparin and heparan sulfate (HS) are structurally similar glycosaminoglycans (GAGs) synthesized from a shared precursor polymer, *N*-acetyl heparosan, with the repeating disaccharide structure [4GlcA β1,4GlcNAc α1,]_n_, also known as the *Escherichia coli* (*E. coli*) K5 polysaccharide. Only about one-third of unfractionated heparin chains contain the AT-binding pentasaccharide sequence, which is critical for its anticoagulant activity. This precursor undergoes several enzymatic modifications, including *N*-deacetylation/*N*-sulfation, the epimerization of GlcA to IdoA, and O-sulfation, resulting in the formation of mature heparin or HS. Notably, heparosan, an acetylated polysaccharide produced by bacteria, lacks these sulfation modifications, making it distinct from heparin. Nevertheless, heparosan can be chemically and chemoenzymatically modified to serve as a valuable precursor for heparin production [36,55]. The unique biological activities of heparin arise from its higher degree and specific patterns of O- and *N*-sulfation compared to HS [46,56].

Heparin is predominantly stored in the secretory granules of mast cells, whereas heparan sulfate is broadly expressed on cell surfaces and within the extracellular matrix [46,57]. The anticoagulant activity of heparin is primarily due to a specific pentasaccharide sequence that binds with high affinity to AT, significantly enhancing its inhibitory effect on factor Xa and thrombin [46,57,58].

This bioactive pentasaccharide sequence, characterized by a precise arrangement and the sulfation of its sugar residues, is critical for its interaction with AT [58] (Figure 2). LMWHs are derived from UFH through controlled depolymerization processes, producing shorter chain fragments with distinct pharmacokinetic and pharmacodynamic profiles. UFH refers to the original heparin preparation with a wide range of molecular weights. LMWHs maintain the antithrombotic efficacy of UFH while offering advantages such as a reduced incidence of heparin-induced thrombocytopenia (HIT) and more predictable bioavailability and dosing regimens [5,55]. The clinical benefits of LMWHs are attributed to their structural differences from UFH, including shorter chain lengths and modified sulfation patterns.

The various therapeutic effects of heparin are almost exclusively due to its interactions, largely electrostatic in nature, with a wide variety of proteins that typically bind to HS. Heparin can either substitute for HS, such as in growth factor action, or displace a protein ligand from immobilized HS-proteoglycans, as seen on endothelial cell surfaces.

While HS exhibits significant structural heterogeneity across different cells and tissues to accommodate preferential binding to specific protein ligands, heparin is relatively more uniform in structure. It is a highly sulfated polymer capable of interacting with a broad range of HS-binding proteins. The extensive sulfation of heparin exposes a variety of binding sequences that are more distinct and varied in HS but often masked in heparin by redundant sulfate groups. Unique binding sequences in heparin, such as those involving rare substituents like the 3-O-sulfated GlcNS residue, are crucial for its specific biological activities. This structural composition enables heparin to engage in diverse biological interactions, contributing to its broad therapeutic potential beyond anticoagulation, including anti-inflammatory, antiproliferative, and antiviral effects.

## 3. How Heparin Works as an Anticoagulant?

Heparin serves as a vital anticoagulant by augmenting the activity of AT, a natural inhibitor crucial in our clotting system. This enhancement occurs through a specific sequence of sugars in heparin, triggering a substantial conformational change in AT that boosts its effectiveness in inhibiting thrombin (factor IIa) and factor Xa (FXa) by nearly a thousand-fold [9,29]. Upon binding to AT, heparin facilitates the exposure of a specialized site on AT that selectively targets and inhibits FXa [59,60]. The presence of calcium ions further amplifies this process by facilitating the formation of complexes with longer heparin chains, thereby enhancing inhibition [61,62,63]. The inhibition of thrombin by heparin requires interactions with longer chains, typically consisting of at least seventeen sugar units [42]. This mechanism elucidates why different forms of heparin, such as LMWHs, exhibit varying effects on FXa and thrombin (Figure 3). Additionally, heparin affects factor IXa (FIXa), with calcium ions also augmenting this interaction [64,65,66].

In addition to these effects, heparin moderately enhances the ability of AT to inhibit other factors like FXIa, FXIIa, and kallikrein through a bridging mechanism [67]. Longer heparin chains make this inhibition stronger [68]. AT can also weakly inhibit factor VIIa (FVIIa) and the tissue factor (TF)/FVIIa complex, with heparin enhancing this effect through a mechanism similar to its action on thrombin and FIXa [68,69]. Overall, the finely tuned anticoagulant action of heparin helps prevent abnormal blood clots while maintaining normal blood flow. By selectively enhancing AT’s inhibition of key factors in the clotting process, heparin ensures that the clotting system is precisely regulated. This balance is crucial in preventing thrombosis and highlights the essential role of heparin in clinical anticoagulation therapy.

## 4. Synthesis and Modifications

Innovative approaches in heparin research have led to significant advancements in synthesis and modification techniques, enabling the production of structurally defined heparin derivatives with tailored properties for diverse biomedical applications. This section explores key methodologies, including chemical synthesis, chemoenzymatic strategies, and advances in the bioengineering modifications of heparin and HS.

### 4.1. Chemical Synthesis

Chemical synthesis offers precise control over the structure and properties of heparin derivatives by assembling monomeric building blocks through stepwise reactions. Early chemical synthesis efforts focused on synthesizing the oligosaccharide fragments of heparin, which served as key intermediates for assembling larger polysaccharide chains [70]. Over the years, advancements in protecting group chemistry, glycosylation reactions, and purification techniques have enabled the synthesis of structurally defined heparin oligosaccharides with specific sulfation patterns and chain lengths [40,71,72,73]. One notable approach is the “block synthesis” strategy, wherein monosaccharide building blocks are sequentially assembled to construct heparin oligosaccharides with defined sequences and sulfation patterns [74,75]. Chemical methods such as glycosylation reactions, selective deprotection, and sulfation steps are carefully orchestrated to ensure regioselective and stereoselective control over bond formation and modification sites. Moreover, solid-phase synthesis techniques have been employed to facilitate the stepwise assembly of heparin oligosaccharides on a solid support, allowing for the efficient purification and characterization of intermediates. Recently, Ramadan and colleagues introduced an automated, machine-assisted solid-phase technique that greatly accelerated the assembly of HS disaccharides [76].

Despite these advancements, the chemical synthesis of heparin remains challenging due to the complexity of its structure, stereochemistry, and regiochemistry. Moreover, scalability and cost-effectiveness issues limit the widespread application of chemical synthesis for the large-scale production of heparin derivatives.

### 4.2. Chemoenzymatic Synthesis

The process of synthesizing heparin chemically presents significant obstacles, including the need for numerous protection and synthesis steps, lower yield, and higher expenses, which hinder its widespread use. To overcome these challenges, scientists have embraced chemoenzymatic synthesis methodologies [77,78]. For example, the capsular polysaccharide extracted from the *E. coli* K5 strain, termed K5 polysaccharide, possessing a repetitive disaccharide composition [4GlcA β1,4GlcNAc α1,]_n_, serves as a precursor for heparin/HS creation [79]. Employing chemoenzymatic synthesis, K5 polysaccharides can be converted into “neoheparin” [79]. Initially, the polysaccharide, also known as heparosan, undergoes chemical *N*-deacetylation and *N*-sulfation. Subsequent enzymatic treatment with C5-epimerase introduces IdoA residues, followed by the selective O-sulfation of hydroxyl groups on both HexA and GlcNS residues [80,81]. Despite yielding products with heparin-like anticoagulant activity, non-selective chemical O-sulfation results in the formation of undesired sulfation at C-3 of HexA, a structure absent in natural heparin and HS. To address this challenge, alternative approaches employing O-sulfotransferases have been devised, significantly enhancing selective O-sulfation reactions [82,83,84]. Notably, the mutational (Mt) expression of heparan sulfate 6-O-sulfotransferase (Hs6st), particularly Hs6st Mt-4, enables the specific sulfation of non-reducing terminal glucosamine residues. Additionally, *N*-deacetylation through chemical processes, coupled with the enzymatic modification of human heparan sulfate 3-O-sulfotransferase (Hs3st1), yields single-site sulfated heparosan, termed as 2-deacetyl-3-O-sulfo-heparosan, which demonstrates promising anti-tumor activity in vitro [85]. Most recently, a scalable method for producing bioengineered heparin identical to porcine heparin has been developed using biosynthetic enzymes on an inert support and requiring *N*-sulfoheparosan with matching *N*-sulfation levels. This heparin can be converted into LMWH resembling USP enoxaparin [86]. Despite promising advancements, achieving industrial-scale production remains a significant challenge in the field of the chemoenzymatic synthesis of heparin.

### 4.3. Advances in Bioengineering

The advancement in heparin oligosaccharides and polysaccharides through sophisticated chemical and chemoenzymatic synthesis techniques presents a highly promising alternative to traditionally sourced heparin from animal and bacterial origins [87,88,89]. However, despite their potential, the existing production capacity of these techniques fails to meet the global demand for heparin active pharmaceutical ingredients (APIs). Given this challenge, there is a growing interest in engineering custom cell factories capable of manufacturing non-animal-derived heparin analogs, leveraging advancements in synthetic biology. While substantial progress has been achieved in producing K5 heparosan [28], a precursor to heparin, limited attention has been paid to developing modified heparin analogs with altered structures and anticoagulant properties.

Notably, investigations have explored the utility of Chinese hamster ovary (CHO) cells and yeast platforms for this purpose. CHO cells are particularly favored due to their innate heparin/HS modification enzymes, rendering them suitable hosts for biosynthesis. Augmenting the expression of specific enzymes, such as Hs3st1 and *N*-deacetylase/*N*-sulfotransferase 2 (Ndst-2), in CHO cells has resulted in the production of HS with improved anticoagulant efficacy, albeit inferior to pharmaceutical heparin [90,91]. Furthermore, efforts have focused on enhancing enzyme localization within CHO cells to optimize activity [92]. Yeast, characterized by its cost-effectiveness and ease of cultivation, emerges as an appealing alternative to mammalian cells for heparin production. Through the genetic manipulation of *Pichia pastoris*, a methylotrophic yeast strain, and the establishment of a cell-free enzymatic system, significant strides have been taken in generating heparin analogs with properties akin to those derived from animals [93]. Additionally, the CRISPR (clustered regularly interspaced short palindromic repeats)/Cas9 technology facilitates the regulation of enzyme expression in the heparin biosynthetic pathway through efficient genome editing [94]. As researchers continue to optimize the expression of the remaining heparin biosynthetic enzymes, the prospect of the large-scale production of metabolically engineered heparin using mammalian cells grows closer (Figure 4) [15,89,91,92,93,95,96].

Overcoming the current obstacles in enzyme arrangement and cooperation within the cellular environment is essential for realizing this goal. Nevertheless, ongoing progress in synthetic biology offers hope for overcoming these challenges. With a deeper understanding of enzyme regulation mechanisms and collaborative efforts, the future establishment of cell factories dedicated to mass-producing heparin appears increasingly feasible. Ultimately, this advancement would represent a transformative achievement in bioengineering heparin, revolutionizing the model of anticoagulant production, and enhancing patient care worldwide.

## 5. Commonly Used Heparin and Its Derivatives

Introducing an extensive variety of therapeutic options, heparin has been segmented into three distinct molecular weight categories—UFH, LMWH, and ultra-low-molecular-weight heparin (ULMWH)—all sanctioned by the FDA [2,5]. UFH, originating from animal tissues like the porcine intestine, poses challenges due to impurities and complex pharmacokinetics, often leading to concerns such as HIT [83]. However, its long-standing history and broad application continue to make it a vital therapeutic option in various clinical scenarios. In contrast, LMWH, derived from UFH through partial depolymerization [40], offers significant advantages such as improved bioavailability and reduced side effects, thereby enhancing patient care and treatment outcomes [5]. Innovations in LMWH production, incorporating novel techniques such as photodepolymerization and chemoenzymatic synthesis, aim to optimize yield and quality while mitigating costs, thereby addressing key challenges associated with traditional manufacturing processes [40,97,98,99]. The evolution of heparin therapy further advances with ULMWH, represented by Fondaparinux, a synthetic pentasaccharide with precise anticoagulant activity targeting factor Xa [100,101]. Despite its enhanced safety profile, ULMWH faces challenges related to production costs and complexity [102]. A comprehensive overview of LMWHs and ULMWHs has been depicted in Table 1. However, ongoing advancements in synthesis methodologies [102,103,104] hold promise for overcoming these obstacles and expanding the clinical utility of ULMWH. These significant developments indicate a transformative future for anticoagulation therapy, in which a wide range of heparin formulations will provide patients with tailored treatment options while driving advances in pharmaceutical manufacturing and drug research.

## 6. Diverse Applications of Heparin

### 6.1. Heparin in Anti-Inflammatory Therapies

Heparin, long recognized for its role as an anticoagulant, has recently garnered attention for its remarkable anti-inflammatory properties. Inflammation plays a crucial role in various diseases, and recent research has elucidated the multidimensional role of heparin in modulating inflammatory pathways. Extensive reviews have shed light on how heparin interacts with key proteins involved in inflammation, including those within the complement system, selectins, and chemokines [2,38,105,106,107,108,109]. These interactions not only inhibit neutrophil activation and vascular smooth muscle cell proliferation but also modulate the expression of inflammatory mediators by engaging with vascular endothelial cells [110]. Beyond its traditional use as an anticoagulant, heparin demonstrates promising anti-inflammatory effects across diverse conditions such as arthritis, inflammatory bowel disease, and bronchial asthma [107,111,112,113,114,115,116,117,118,119,120,121]. Exciting developments include novel formulations such as LMWH MMX (CB-01-05-MMX), a novel oral Parnaparin sodium, which shows potential in treating ulcerative colitis with favorable safety profiles observed in clinical trials [111]. Moreover, inhaled heparin emerges as a potential adjunctive therapy for chronic obstructive pulmonary disease and asthma, offering both anti-inflammatory and mucolytic benefits [117,122]. In acute pancreatitis cases characterized by elevated triglyceride levels, combination therapy involving LMWH calcium and insulin has demonstrated efficacy in improving immune function and coagulation parameters [118]. Sepsis, a severe and life-threatening condition triggered by Gram-negative bacteria, has drawn intense scrutiny due to its high mortality rate [1]. The interactions of heparin with pro-inflammatory factors and involvement in procoagulant cascades hold promise in mitigating the inflammation and coagulopathy associated with sepsis [107,123,124].

Recent investigations underline the potential of heparin to disrupt high mobility group box 1 protein–lipopolysaccharide interactions, yielding significant anti-inflammatory effects [125,126]. Additionally, heparin and selectively desulfated heparin have shown efficacy in binding to positively charged histones, thereby attenuating their inflammatory actions [127,128]. Recent findings suggest that the altered forms of heparin derivatives possess the ability to bind to positively charged histones, thereby mitigating histone-induced inflammation [129]. Specifically, selectively desulfated heparin, with diminished anticoagulant properties, retains its effectiveness as an anti-histone agent. For instance, a derivative known as anti-thrombin affinity-depleted heparin (AADH), lacking anticoagulant activity, directly binds to histones, reducing histone-mediated cytotoxicity and mortality in inflammation and sepsis models in mice without heightening bleeding risk [129]. Moreover, during sepsis, the oligosaccharides of heparin or HS may influence patient cognitive functions [130]. Therapeutically, doses of UFH protect the glycocalyx from shedding by reducing inflammation in models of septic shock. In treating sepsis, LMWH serves as a vital component of supportive therapy [124,131]. It helps improve multiple organ dysfunction syndrome and reduces the mortality rate within a shorter period. The recent assessments of the effectiveness of UFH highlight a notable decrease in mortality rates over 28 days, accompanied by minimal bleeding risks [132]. Unlike LMWH, UFH provides a spectrum of effects beyond its anticoagulant properties. It can disrupt nuclear factor NF-κB activation, block the movement of chemokines and monocytes, bolster the endothelial barrier, and regulate the angiopoietin (Ang)/Tie2 axis [133]. Although the anti-inflammatory properties of heparin are separate from its blood-thinning effects, concerns about bleeding limit its use as an anti-inflammatory treatment. Therefore, refining heparin to minimize or eradicate its anticoagulant activity holds promise in enhancing its efficacy against inflammatory diseases.

### 6.2. Heparin in COVID-19 and Other Infectious Diseases

Exploring the potential of heparin as a COVID-19 treatment, particularly in severe cases with clotting issues and organ damage, has sparked interest [2,28]. Recent research suggests that heparin may effectively target multiple aspects of COVID-19, including thrombosis prevention, the inhibition of viral entry, and reduction in cytokine activity, supported by both clinical observations and experimental data [5]. The presence of coagulation abnormalities in COVID-19 poses an ominous threat, often leading to high mortality rates; however, studies suggest that administering UFH or LMWH could potentially reduce mortality in critically ill patients by preventing blood clot formation [23,134,135,136,137]. The initial recognition of the anticoagulant properties of heparin in COVID-19 stemmed from a retrospective study that included nearly 500 patients in Wuhan, China [138]. Subsequent studies across the US and Italy further supported the link between heparin usage and reduced mortality rates, particularly among elderly patients [136,139]. These findings emphasize the importance of early prophylactic anticoagulation upon hospital admission for COVID-19 patients, as recommended by the International Society for Thrombosis and Hemostasis (ISTH) guidelines [140].

Apart from its function as an anticoagulant, heparin exhibits direct antiviral effects by attaching to SARS-CoV-2 proteins, impeding viral attachment and replication [141,142,143]. This interference with the virus–host cell interaction suggests heparin’s potential in fighting COVID-19 and similar coronaviruses. Additionally, heparin displays encouraging signs in blocking the main protease of SARS-CoV-2, a key process in viral replication [144]. These findings highlight the potential of heparin as a first-line antiviral treatment for COVID-19 [142]. Moreover, heparin shows positive effects in reducing inflammation linked to COVID-19, particularly in the cases of cytokine storm syndrome (CSS) observed in severe infections [145]. By interacting with inflammatory cytokines and pathways, heparin could potentially alleviate the systemic inflammatory response often seen in severe COVID-19 cases. Additionally, there is promising evidence suggesting that heparin may be effective in managing COVID-19-associated myocarditis, thereby broadening its therapeutic utility [146]. Sun and colleagues observed a significant increase in a protein called heparin-binding protein (HBP) in severely ill COVID-19 patients as their condition worsened. This observation emphasizes the potential involvement of HBP in the systemic inflammatory response associated with severe COVID-19. The correlation between HBP levels and the deterioration of COVID-19 symptoms suggests that HBP may serve as a crucial disease indicator [147]. Furthermore, recognizing HBP as a potential therapeutic target in COVID-19 underscores its significance in developing targeted treatment strategies aimed at mitigating the severe inflammatory response observed in critically ill patients. In summary, heparin provides a wide array of benefits for COVID-19 patients, serving as both an anticoagulant, antiviral, and anti-inflammatory treatment. However, to fully harness its potential, further research is needed to fine-tune the dosage, timing, and duration of heparin administration, especially for patients with unique medical profiles [27]. As researchers delve deeper into the therapeutic potential of heparin, it becomes increasingly clear that it holds significant promise in combating COVID-19.

In addition to ongoing research into COVID-19, heparin and its derivatives exhibit promising abilities to hinder the adhesion and invasion of various viruses such as dengue virus (DENV), herpes simplex virus (HSV), human immunodeficiency virus (HIV), influenza virus, Zika virus, and human papillomavirus 16 (HPV16) [148,149,150,151,152,153,154,155,156,157]. Heparin also directly impacts viruses and bacteria, reducing iron levels in human macrophages to control *Mycobacterium tuberculosis* bacterial replication and partially impeding Zika virus replication [155,158]. This dual action prevents virus-induced cell necrosis and apoptosis without harming uninfected cells, while simultaneously activating cell survival signaling pathways [159].

Malaria, a significant infectious epidemic, demonstrates the antimalarial potential of heparin as it specifically binds to Plasmodium-infected red blood cells (pRBCs) and liver circumsporozoites [160,161,162]. The nanomedical applications of heparin have emerged to transport drugs to the mosquito stages of malaria parasites [16]. A recent study introduces dendronized hyperbranched polymers (DHPs) designed for loading antimalarial agents and coated with heparin to target red blood cells infected with *Plasmodium falciparum*. These DHP–heparin complexes exhibit both the inherent antimalarial properties of heparin, with an IC50 of around 400 nM, and specific targeting towards *P. falciparum*-infected erythrocytes compared to the uninfected ones. This innovative approach presents a promising addition to the limited repertoire of structures available for effectively loading and delivering antimalarial agents [163]. In Lyme disease (LD), heparin derivatives such as non-anticoagulant heparin (NACH) impede *Borrelia burgdorferi* sensu lato (Bbsl) attachment to mammalian cells and boost antibody immune responses to thwart LD [164,165]. Moreover, exploiting the natural attraction between bacteria or viruses and heparin offers promising avenues for capturing and eliminating them from circulation. This strategy is particularly useful in extracorporeal medical devices such as the Seraph^®^ 100 Microbind^®^ Affinity Blood Filter [166]. This pioneering device integrates polyethylene beads immobilized with heparin, authorized for reducing and eradicating pathogens from the bloodstream. Preliminary trials demonstrate its efficacy in diminishing viral loads, including Zika virus, adenovirus, and SARS-CoV-2, indicating its potential as an adjunct therapy for severely ill COVID-19 patients [26,167].

### 6.3. Heparin in Oncology

Since 1865, when Armand Trousseau linked superficial migratory thrombophlebitis (SMT) to an underlying malignancy, the intricate relationship between cancer and thrombosis has been acknowledged [168,169]. Among cancer-assisted thrombotic events, venous thromboembolism (VTE) including deep vein thrombosis (DVT) and pulmonary embolism (PE) reigns supreme, often necessitating treatment with LMWH [170,171,172]. Recent insights assembled from reviews and meta-analyses suggest that heparin and its derivatives extend beyond their anticoagulant role, exerting a direct influence on tumor biology pathways, including inflammation, angiogenesis, and metastasis [173,174]. One notable target in cancer therapy is heparanase, an endoglycosidase prevalent in malignant tumors [175,176,177]. Heparin and its mimetics show promise in selectively inhibiting heparanase expression while also targeting essential mediators such as fibroblast growth factors (FGFs) and the vascular endothelial growth factor (VEGF) involved in tumor angiogenesis [173,178,179,180]. Furthermore, modified heparin derivatives demonstrate potential in blocking galectin 3-mediated cancer cell adhesion and angiogenesis, offering a new frontier in anti-metastasis and anti-cancer drug development [181].

Despite encouraging preclinical data, clinical trials evaluating the efficacy of heparin and its derivatives in cancer treatment encounter obstacles [182]. For instance, in lung cancer phase III trials, LMWH failed to significantly improve survival rates [183,184]. Similarly, the non-anticoagulant heparin analog necuparanib did not achieve the anticipated efficacy in phase II clinical trials for pancreatic cancer, highlighting the challenges in developing effective non-anticoagulant heparin-based therapeutics for oncology applications [185]. Moreover, adverse effects, including bleeding-related complications, have been observed with heparin mimetics such as PI-88 [182]. Nonetheless, ongoing research ventures into novel heparin mimetics such as PG545 (pixatimod), which boasts immunomodulatory and antiangiogenic properties [186]. Phase I trials have exhibited promising safety profiles and encouraging disease progression control, particularly in advanced solid tumors [187]. With PG545 now undergoing phase II trials in combination with nivolumab [186], further exploration is warranted to unlock the potential of heparin and its derivatives in cancer treatment.

Subcutaneous LMWH administration has limitations, including adverse reactions at the injection site, discomfort, bruising, and bleeding, which can significantly affect patients’ quality of life [188]. However, recent advancements have led to the emergence of direct oral anticoagulants (DOACs) as a viable alternative for treating cancer-associated thrombosis in specific patient cohorts. Supported by the results of numerous high-quality randomized controlled trials (RCTs), DOACs have gained recognition as an effective and safe option, receiving strong recommendations in clinical guidelines [189,190,191,192,193,194]. Chronologically, these findings have progressed alongside the understanding of thromboprophylaxis after cancer-related surgery. While DOACs are primarily aimed at preventing thrombosis, they presumably lack the other potentially valuable effects of heparin-based drugs of a polyanionic nature. Specifically, DOACs, such as apixaban and rivaroxaban, have demonstrated equivalence to LMWH in preventing postoperative venous thromboembolism (VTE). Consequently, oral DOACs are increasingly considered as potentially effective and safe alternatives to subcutaneous LMWH for thromboprophylaxis in patients undergoing cancer surgery [195].

Several methods are available to address the neutralization of overdosed heparin-based anticoagulants, each with its own limitations. Protamine sulfate is the primary agent used to neutralize UFH by forming a stable complex with heparin, effectively reversing its anticoagulant effects. However, when used for LMWHs, protamine only partially neutralizes their activity, typically reversing about 60–80%, leaving residual anticoagulant effects that can pose a clinical challenge, especially in cases of significant overdose. Additionally, protamine sulfate can cause adverse reactions, such as hypotension, bradycardia, and anaphylaxis, particularly in patients with fish allergies or previous exposure to protamine. Alternative approaches, such as hemodialysis or the use of agents like recombinant factor VIIa or activated prothrombin complex concentrates, have been explored for reversing heparin effects, but they are less effective and carry their own risks. These limitations underscore the complexity of managing heparin overdoses and highlight the need for careful monitoring and individualized treatment plans to mitigate the risks associated with both heparin use and its neutralization [196].

Above all, while evidence suggests the promising role of heparin and its derivatives in cancer therapy, a deeper understanding of their mechanisms of action is imperative [173]. Furthermore, the innovation of heparin mimetics with diminished anticoagulant properties, coupled with the refinement of dosing protocols, is imperative for improving therapeutic effectiveness and safety [5,197]. Extensive clinical investigations into their anti-cancer effects, both standalone and in combination with the existing therapies, offer hope for advancements in cancer treatment.

### 6.4. Heparin in Nephropathy

In renal medicine, heparin holds a vital position as a potent anticoagulant, valued for its efficacy in preventing blood clot formation and mitigating thrombotic risks. Its role extends across diverse renal conditions, including nephrotic syndrome, diabetic nephropathy, and hemodialysis, where it serves as a reliable ally in patient management. Nephrotic syndrome, a condition characterized by the injury of vascular endothelial cells, triggers a hypercoagulable state in the body, increasing the risk of complications such as lower limb venous thrombosis, particularly prevalent in children [198]. Clinical management often involves a comprehensive approach encompassing anticoagulants, anti-infective, and vasodilator drugs [198,199,200,201,202]. Among these, LMWH emerges as a prominent therapeutic option, exhibiting robust anticoagulant properties. LMWH effectively mitigates secondary pathological hypercoagulability by augmenting blood viscosity and inhibiting kidney blood clot formation, thus minimizing the likelihood of thrombosis-related complications and bleeding incidents [201]. Additionally, LMWH demonstrates anti-inflammatory and antiproliferative effects, fostering extracellular matrix repair and endothelial cell restoration, ultimately alleviating kidney damage. This treatment modality holds promise in addressing the complex interplay of coagulation abnormalities and renal dysfunction, offering significant therapeutic potential in the management of nephrotic syndrome [202].

Diabetic nephropathy (DN) represents a significant challenge among individuals with diabetes, posing a substantial burden on health outcomes. Recent clinical investigations highlight the promising role of sulodexide, an LMW GAG that is actually a mixture of two GAGs—HS and dermatan sulfate (DS)—in ameliorating proteinuria in DN patients, even when co-administered with angiotensin-converting enzyme (ACE) inhibitors or angiotensin receptor antagonists, suggesting its potential for renal protection [203,204]. The proposed mechanism of sulodexide involves the inhibition of heparanase activity, thus preserving the integrity of HS within the glomerular capillary wall and restoring the ionic permselectivity of the glomerular basement membrane (GBM) [205]. Ongoing clinical trials seek to further elucidate the renal protective properties of sulodexide in DN [206]. Additionally, the exogenous administration of GAGs induces notable chemical and anatomical alterations in renal tissues, with both heparin and LMWH exhibiting suppressive effects on inflammatory responses within diabetic glomeruli [207,208]. Experimental investigations in rodent models indicate that altered heparin with decreased anticoagulant efficacy efficiently alleviates glomerular and tubular matrix deposition, suppresses the expression of transforming growth factor beta (TGF-β1), and diminishes albuminuria. These findings imply a promising prospect for GAG therapy in thwarting diabetic glomerulosclerosis by mitigating TGF-β1 overexpression [209]. The current therapeutic strategies for DN emphasize the identification of pivotal molecular targets, with emerging evidence suggesting a correlation between kidney injury and receptor for advanced glycation end products (RAGE) gene expression. LMWH emerges as a promising RAGE antagonist, exerting notable renoprotective effects by alleviating albuminuria, augmenting glomerular cell count, and attenuating mesangial expansion, thus highlighting the therapeutic promise of RAGE antagonists in DN management [210,211].

In the field of allergic purpura nephritis among pediatric patients [212], LMWH calcium stands out as a crucial treatment option. Its therapeutic significance extends beyond its anticoagulant properties, encompassing the prevention of thrombosis, modulation of blood viscosity, and facilitation of basement membrane reconstruction [213]. By addressing these diverse aspects, LMWH plays a vital role in alleviating the severity of nephritis in the affected children. These observations underscore the various functions of LMWH in treating kidney diseases, suggesting a hopeful path for improving patient outcomes in different clinical situations.

Currently, the practice of hemodialysis is widely used in clinical settings to prevent acute renal failure. Nonetheless, ensuring uninterrupted treatment necessitates the prevention of blood coagulation [214]. Although heparin is commonly employed to inhibit clot formation in hemodialysis apparatus, its efficacy in minimizing the risk of bleeding during hemodialysis remains limited. In response, LMWH has gained widespread adoption in clinical practice to mitigate this concern. LMWH serves to exert an anticoagulant effect, thus reducing the likelihood of bleeding events in patients undergoing hemodialysis, while also boasting an extended duration of action [215]. In summary, heparin, particularly LMWH, plays a critical role in the management of renal diseases by preventing thrombosis and reducing the risk of thrombotic complications. Its use is guided by the specific renal condition and individual patient factors, highlighting the importance of personalized treatment approaches in renal care.

### 6.5. Heparin in Cardiopathy

Heparin serves as a cornerstone in the management of acute coronary syndrome (ACS), a critical condition characterized by acute myocardial ischemia due to thrombosis triggered by unbalanced atherosclerotic plaque rupture or erosion within the coronary arteries [216]. ACS encompasses a spectrum of events, including ST-segment elevation myocardial infarction (STEMI), non-ST-segment elevation myocardial infarction (NSTEMI), and unstable angina (UA), all with the potential to progress to acute myocardial infarction or sudden cardiac death.

During the acute phase of ACS, primary percutaneous coronary intervention (PCI) stands as the preferred approach for treating STEMI, while intravenous thrombolysis offers an alternative for patients ineligible for immediate PCI [217]. Thrombolysis, commonly utilizing urokinase or streptokinase, is complemented by anticoagulant therapy, typically involving UFH or LMWH [218]. LMWH, exemplified by enoxaparin, has emerged as a favored option, offering comparable or superior efficacy to UFH with a more favorable safety profile. In the management of coronary heart disease (CHD) with ACS, anticoagulant therapy plays a central role in preventing thrombotic complications and optimizing outcomes. Heparin acts as an adjunct to both PCI and thrombolytic therapy in STEMI cases and forms a cornerstone in early conservative strategies for NSTEMI [219,220]. During PCI procedures, intravenous anticoagulation with either heparin or bivalirudin, coupled with antiplatelet agents, is standard practice [221,222,223,224]. Bivalirudin is a synthetic direct thrombin inhibitor used as an anticoagulant, primarily in patients undergoing PCI for ACS, particularly STEMI. While bivalirudin exhibits similar efficacy and safety to heparin, the latter offers substantial cost advantages [225,226]. In a recent examination of randomized controlled trials, Al-Abdouh et al. conducted a systematic review and meta-analysis, which included the landmark BRIGHT-4 (bivalirudin with prolonged full-dose infusion during primary PCI vs. heparin) trial. Their findings revealed that bivalirudin, administered during PCI for MI, resulted in reduced rates of major bleeding and cardiovascular mortality compared with unfractionated heparin. Surprisingly, there were no significant differences observed in major adverse cardiovascular events, all-cause mortality, MI, stroke, or stent thrombosis between the two treatment groups [227]. These results closely correspond with the outcomes of several other meta-analyses conducted on similar subjects [228]. Furthermore, LMWH, particularly enoxaparin, has captured interest as a preferred post-PCI anticoagulant owing to its superior efficacy and safety compared to UFH, eliminating the need for routine anticoagulant intensity monitoring [224,229]. Emerging evidence advocates for the use of LMWH-heparin combination therapy in complex coronary artery disease cases undergoing PCI, highlighting the evolving landscape of anticoagulation strategies in the management of CHD.

### 6.6. Heparin in Neuroprotection

Heparin has demonstrated promising neuroprotective effects in neurodegenerative diseases and traumatic brain injury by modulating neuroinflammation, promoting neuronal survival, and enhancing neuroregeneration [230,231]. This multifaceted mechanism of action highlights its potential as a therapeutic intervention for conditions such as Alzheimer’s disease (AD), Parkinson’s disease (PD), and stroke. AD poses a significant burden on elderly healthcare systems worldwide, characterized by cognitive decline and dementia, leading to diminished quality of life and increased mortality rates among the affected individuals. The development of AD is characterized by the gradual buildup of amyloid plaques and neurofibrillary tangles, which are considered the hallmark features of the condition [232]. This accumulation occurs progressively within the brain, representing a slow and irreversible process that poses significant challenges for therapeutic interventions. These pathological features not only underscore the complexity of AD but also impede the development of effective treatments.

Recent research has elucidated the role of HS proteoglycans in AD pathogenesis [233], influencing the formation of plaques and tangles while facilitating pathogen entry into brain cells. This understanding has prompted exploration into the therapeutic potential of heparin and its derivatives in AD management. Studies have demonstrated their capacity to mitigate various aspects of AD pathology, including the reduction in amyloid peptide levels, inhibition of tau phosphorylation, and attenuation of inflammatory responses [230,231]. Notably, certain LMWHs have shown promise in reducing beta-amyloid plaque accumulation and enhancing cognitive function in AD animal models [234]. Additionally, heparin oligosaccharides have exhibited neuroprotective properties by modulating amyloid precursor protein secretion [235] and impeding the uptake of tau aggregates, offering novel avenues for AD therapy [236]. Despite the challenges posed by the anticoagulant activity of heparin, researchers are investigating marine-derived heparin analogs with enhanced therapeutic profiles, targeting key enzymes implicated in AD pathology without significant anticoagulant effects [237,238,239]. Moreover, synthetic heparin oligosaccharides such as SN7–13 have shown promise in blocking tau protein uptake and inhibiting tau aggregate formation, presenting innovative strategies for addressing tauopathies associated with AD [205]. Recent research findings propose that the increased expression of the HS3ST1 gene might facilitate the propagation of tau pathology, revealing a novel target for potential therapeutic interventions in AD [240]. On the other hand, another neurodegenerative disorder, PD, is marked by the progressive loss of dopaminergic neurons situated in the substantia nigra area of the brain. This loss results in evident clinical symptoms, such as involuntary tremors during periods of rest, sluggish movement (bradykinesia), and challenges in maintaining normal posture and walking [241]. In a recent study conducted by Wang and his team, they examined the effectiveness of LMWH and LMWCS in treating PD. The research found that LMWH helped alleviate mitochondrial dysfunction due to its antioxidant properties, while LMWCS was effective in reducing neuroinflammation by blocking platelet activation [242]. As research progresses, heparin-based therapies continue to emerge as promising modalities for addressing the adaptable challenges posed by AD, PD, and related neurodegenerative diseases.

### 6.7. Heparin in Nanomedical Research and Drug Delivery Systems

In recent years, the landscape of nanomedicine has been transformed by the remarkable integration of heparin, once solely known for its anticoagulant properties. Its journey from an anticoagulant to a multi-dimensional player in the nanotechnology arena has been propelled by its unique attributes, including biocompatibility, biodegradability, and intricate interactions with biological molecules. This transformation has steered us into a new era, where the utilization of heparin as a coating material for nanoparticles has become particularly noteworthy, enriching their stability and biocompatibility for a diverse array of biomedical applications. Additionally, the advent of heparin-functionalized nanoparticles has heralded a promising avenue for precise drug delivery, offering targeted treatment with minimized off-target effects and systemic toxicity [51,52]. Moreover, the immunomodulatory properties of heparin have attracted attention in nanomedicine, particularly in inflammation-associated diseases such as cancer, cardiovascular diseases, and autoimmune disorders. By facilitating the delivery of anti-inflammatory agents to targeted sites, heparin-coated nanoparticles offer the potential to effectively attenuate inflammatory responses [49,50].

In addition, advancements in polymer-based nanocomposites have driven progress in biomedical applications, with heparin playing a central role [40,53]. Serving as a highly bioactive polymer, heparin enhances the biotic competence of nanocomposites, expanding their utility across diverse clinical scenarios. The incorporation of the structural and chemical derivatives of heparin enables the fabrication of resourceful nanocomposites tailored for specific clinical applications, including drug delivery, wound healing, tissue engineering, and biosensing [49,50]. Recent developments in heparin-oriented nanotechnology present promising avenues for advancing healthcare delivery and treatment modalities. This section underlines the successful integration of heparin and its derivatives into nanocomposites, emphasizing their potential in both laboratory research and clinical practice (Figure 5). Understanding the advantages and challenges associated with heparin-based nanocomposites is crucial for fostering future innovations in this dynamic field [50].

#### 6.7.1. Suppressing Cancer Progression with Heparin Nanocomposites

Heparin is recognized for its diverse interactions with the biomolecules involved in tumorigenesis, influencing both tumor development and metastasis. Specifically, LMWHs interact with growth factors such as VEGF (vascular endothelial growth factor) and bFGF (basic fibroblast growth factor), interfering with their activity and inhibiting angiogenesis, thus slowing tumor progression [243]. Furthermore, heparin’s inhibition of cell adhesion molecules such as P-selectins and integrins impacts crucial pathways in cancer advancement, including proliferation, metastasis, invasion, and angiogenesis [244]. Exploiting these properties, various nano-drug delivery systems incorporate heparin and LMWH coatings to effectively combat metastasis [245]. These methods have demonstrated heightened cytotoxicity against breast cancer cells and successful tumor management [246]. Heparin-modified liposomes, particularly when carrying photosensitizer cargo, display enhanced anti-cancer and anti-metastatic effects by impeding platelet adhesion and diminishing migration and invasion in breast cancer models [245,247,248]. Moreover, heparin-modified graphene oxide nanocomposites deeply infiltrate tumors, inducing hypoxia, promoting vascular normalization, and counteracting the overexpression of pro-oncogenic markers [249]. This positions them as potential agents for suppressing both tumor growth and metastasis. Additionally, innovative strategies such as drug-loaded heparin-conjugated graphene oxide masked by cancer cell membranes offer distinct approaches for integrated photothermal and chemotherapeutic immunotherapy against melanoma. In the realm of metastasis, tumor cells exhibiting heightened heparanase expression facilitate epithelial–mesenchymal transition (EMT), a pivotal process for hematogenous metastasis [250]. LMWH-conjugated chlorin e6 in micelles has shown efficacy in inhibiting EMT in breast cancer stem cells, suggesting therapeutic potential in metastasis inhibition.

#### 6.7.2. Targeting Angiogenesis with Heparin-Functionalized Nanoparticles

HS emerges as a key contributor in angiogenesis, navigating through a complicated path affected by chain length and sulfation position. Its dual role as both a promoter and inhibitor of blood vessel formation presents its significance in neovascularization [251]. Within nanocomposites, heparin engages in detailed conversations with angiogenic growth factors (AGFs) such as VEGF and FGF-2, directing endothelial cell responses [54,252]. Interestingly, nanoparticles linked with diaminopyridinyl-derivatized heparin show potential in halting FGF-2-induced angiogenesis, unveiling the therapeutic promise of heparin interventions [54]. Furthermore, heparin-functionalized nanocomposites, such as collagen or hydroxyapatite matrices, carefully regulate VEGF release, encouraging the emergence of new blood vessels [253]. In ischemic models, cryogel-based nanoscaffolds loaded with VEGF and heparin offer promise, while heparin-loaded hydrogels modulate angiogenesis through VEGF and cPGE pathways [254]. Coating nanosheets with LHT7 boosts their anti-tumor effectiveness by suppressing angiogenesis, and heparin nanoparticles exhibit an impressive ability in crossing the blood–brain barrier to hinder glioma proliferation [255]. With surface-functionalized heparin nanoparticles delivering potent anti-cancer drugs, the varied applications of heparin in angiogenesis modulation and cancer therapy take center stage, promising a new period in biomedical innovation.

#### 6.7.3. Tailored Heparin Nanocomposites for Enhanced Regeneration

In tissue regeneration and wound healing, heparin plays a versatile role, expediting the healing process through various mechanisms. Heparin enhances the expression of crucial growth factors such as hepatocyte growth factor (HGF), nitric oxide (NO), granulocyte colony-stimulating factor (G-CSF), and vascular endothelial growth factor (VEGF), promoting endothelial and satellite cell proliferation while regulating myeloperoxidase activity and triggering anti-inflammatory responses at injury sites [256]. Extensive research validates the effectiveness of heparin in burn wound healing, attributed to its anti-inflammatory properties, immune modulation, NO production, and the accumulation of proangiogenic factors [257]. Additionally, heparin-loaded nanofiber sutures and scaffolds show promising results in tissue regeneration, facilitating the repair of Achilles tendons and nerves in animal models [258]. These heparin-functionalized nanofibers not only support cell growth and nerve extension but also serve as excellent wound dressings, especially when combined with antimicrobial or anti-inflammatory agents. Nanocomposites incorporating heparin, such as heparin–polyvinyl alcohol@gold nanocomposite (H-PVA@Au), a highly porous bandage, enhance wound healing by promoting skin and collagen formation while exerting antimicrobial effects [259]. Furthermore, heparin serves as a valuable scaffold material and functionalizing agent in other scaffolds and nanoformulations, improving cell adhesion and overall biocompatibility [260,261]. Customized nanocomposites modified with heparin and growth factors such as recombinant human bone morphogenetic protein-2 (rhBMP-2) demonstrate enhanced bone regeneration capabilities, while heparin-loaded fibrous membranes facilitate mesenchymal stem cell attachment and proliferation [262]. These collective findings draw attention to the diverse applications of heparin in tissue engineering and regenerative medicine, opening avenues for innovative biomedical interventions.

In addition, heparin exhibits remarkable versatility, functioning as both an effective anti-inflammatory agent and a potent antiviral compound, showcasing its broad therapeutic potential across various medical fields. Through its interactions with various immune components, heparin effectively neutralizes inflammatory chemokines and regulates complement factors, thereby curbing inflammatory responses [263]. Moreover, it disrupts leukocyte attachment by engaging with adhesion molecules and interferes with inflammatory transcription pathways [127,264,265]. Recent studies have emphasized its ability to modulate multiple inflammation-related signaling pathways, highlighting its potential as a therapeutic agent for inflammatory conditions [266]. When incorporated into nanocomposite materials such as nanofiber matrices and nanoparticle formulations, heparin demonstrates potent anti-inflammatory properties with minimal side effects, further enhancing its therapeutic potential. In addition to its anti-inflammatory role, heparin exhibits strong antiviral activity against enveloped viruses and those utilizing HS proteoglycans as entry receptors. By impeding viral attachment, competing for coreceptor binding, and disrupting viral replication processes, heparin effectively suppresses viral propagation across a range of viruses [267,268,269]. While its mechanism against SARS-CoV-2 remains unclear, heparin presents promise as a therapeutic agent [270]. Innovative approaches incorporating heparin into liposomes and engineered nanomaterials [271,272] offer exciting prospects for combating viral infections and detecting viral pathogens.

#### 6.7.4. Heparin in Smart Drug Delivery Systems

Heparin-based smart drug carriers have received considerable attention in recent years due to their potential to revolutionize chemotherapy drug delivery. These carriers exploit the unique properties of heparin to enhance drug loading, biocompatibility, and targeted delivery. By chemically linking heparin with hydrophobic drugs such as paclitaxel (PTX) [273] via pH-sensitive cis-aconitic bonds [274], researchers have created polymer drugs that can self-assemble into micelles [274]. These micelles, which have a hydrophobic core and a hydrophilic, negatively charged shell, provide an optimal environment for drug encapsulation and controlled release. The versatility of heparin allows for further functionalization; for example, doxorubicin hydrochloride (DOX·HCl) and cationic folic acid (CFA) can be incorporated through electrostatic interactions, endowing the micelles with both passive and active tumor-targeting capabilities. This dual pH sensitivity ensures that both PTX and DOX are released in the acidic tumor microenvironment, enhancing therapeutic efficacy while minimizing systemic toxicity [275]. Compared to traditional PCL-heparin conjugates [276], these heparin-based carriers offer higher drug loading capacity (DLC) and improved biocompatibility, addressing key challenges in chemotherapy such as drug resistance and side effects [275]. This innovative approach highlights the potential of heparin as a robust platform for the development of multifunctional drug carriers, paving the way for more effective and safer cancer treatments.

## 7. Challenges and Opportunities

(i) Extending half-life for improved compliance: The short half-life of heparin necessitates frequent administration, resulting in poor patient compliance. However, ongoing efforts to develop extended-release formulations aim to address this challenge and potentially improve patient adherence to treatment regimens.

(ii) Extraction challenges and solutions: The complex extraction process of crude heparin from animal sources remains a limitation compared to other biopolymers. Although significant strides have been made in depolymerization, chemical modification, and synthetic production, it is important to recognize that achieving a sustainable alternative to animal-derived heparin still has a long way to go [5,277].

(iii) Management of bleeding risk and prevention of thrombocytopenia: While heparin is highly effective as an anticoagulant, it carries a risk of bleeding, especially in vulnerable populations such as the elderly or those with renal insufficiency [2]. Strategies to monitor and mitigate the risk of bleeding, along with advances in anticoagulant therapy, offer opportunities to improve patient safety during heparin treatment. In addition, HIT is a major concern because of its potential to lead to severe complications, including thrombotic events [83]. Understanding the mechanisms underlying HIT and developing novel approaches to prevent or mitigate its occurrence could significantly improve patient outcomes and safety during heparin therapy.

(iv) Advancing precision medicine approaches: The potential to optimize heparin therapy and reduce the risk of adverse reactions in individual patients can be realized by leveraging advances in personalized medicine, such as pharmacogenomics and biomarker identification. Tailoring treatment regimens based on patient-specific factors and genetic profiles could improve efficacy and safety while reducing the incidence of adverse events associated with heparin treatment.

(v) Environmental sustainability and improved bioavailability: In response to the escalating challenges of environmental pollution and climate change, there is an increasing demand for biodegradable and sustainable materials in nanocomposites. Heparin, acknowledged as a green polymer, offers a promising solution due to its biocompatibility and non-toxic properties. Furthermore, the low bioavailability of heparin and its low molecular weight counterparts, which refers to the limited amount of the drug reaching systemic circulation after administration, presents a significant challenge that can be effectively addressed by incorporating them into nanocomposites [278]. This strategic integration not only improves bioavailability, but also mitigates the generic toxicity of nanomaterials.

(vi) Clinical translation challenges: Despite significant progress, the translation of heparin nanocomposites from preclinical studies to human clinical trials remains challenging [2,5]. Issues such as unpredictable dosing efficiency, target site accumulation, and diffusion hinder their clinical translation [50] and require further research and optimization.

(vii) Monitoring and management of other adverse effects: While some adverse effects of heparin treatment, such as increased eosinophils or hyperkalemia, are relatively rare and reversible with the cessation of treatment, others, such as calcium deposition at injection sites in patients with chronic renal failure, require careful monitoring and management [259]. The development of tailored approaches to address these specific adverse effects could improve the safety and tolerability of heparin therapy in the affected patient population.

## 8. Advancing Heparin Research

The integration of heparin into multiple therapeutic modalities stands as an indication of the persistent pursuit of transformative breakthroughs in medical research, promising to revolutionize precision medicine. By tapping into the diverse properties of heparin, researchers are exploring innovative paths in targeted therapy, wound care, bioimaging, and infection control. Engineered heparins, synthesized through chemical, chemoenzymatic, and metabolic engineering approaches, have emerged as promising alternatives to animal-sourced heparin, addressing concerns over the fragility of the heparin supply chain and the recent contamination incidents [40,47]. These engineered heparins have the potential to fine-tune heparin-binding motifs and other molecular characteristics, thereby enhancing therapeutic efficacy and reducing side effects. However, the development of these alternatives is still ongoing and requires further research to fully realize their potential [279]. For example, microneedle patches delicately infused with heparin exemplify a departure from the traditional systemic anticoagulation methods, offering localized delivery that minimizes systemic side effects and improves patient adherence [279]. This targeted approach not only improves therapeutic efficacy but also alleviates the burden on patients undergoing anticoagulation therapy. Moreover, the synergy between heparin and electroconductive hydrogels presents an enticing avenue for advancing wound healing and nerve regeneration, with the incorporation of the clotting modulation of heparin with the bioactive properties of polymers such as polyaniline and polypyrrole opening new frontiers for therapeutic interventions [279,280]. Compounds that perform similar functions as heparin, such as binding to the heparin-binding site on a protein, may also be characterized as heparin mimetics, and the synthesis of clinically useful heparin mimetics is a relatively recent achievement, with the prospect of developing mimetics that display higher relative potency and greater selectivity of action than their parent molecule being a major driving factor in this field of research [197]. The fusion of heparin with cutting-edge nanotechnology further underscores its versatility, as heparin-based nanoplatforms demonstrate remarkable capabilities in enhancing imaging contrast and combating microbial proliferation, thereby offering a comprehensive solution to pressing medical challenges [49]. As these innovations continue to mature and scale industrially, they hold the promise of not only improving clinical outcomes and elevating patient care standards but also broadening access to advanced therapeutic solutions. With ongoing research and innovation, the journey of heparin in medical science is poised to reveal unique possibilities for personalized and targeted therapies.

## 9. Conclusions and Future Outlook

In summary, beyond its traditional role as an anticoagulant, heparin’s multifaceted utility spans a broad range of medical applications. This review has provided a comprehensive overview of its diverse therapeutic uses and its relevance in nanomedicine. Heparin’s intricate interactions with biological components enable it to regulate fundamental processes critical to both health and disease. From its involvement in inflammation and wound healing to its significance in cancer therapy and infectious disease management, heparin is emerging as a key player in both basic and clinical research.

To advance these applications, several challenges must be addressed, such as managing heparin’s anticoagulant side effects and elucidating its mechanisms and roles in various diseases. Research should focus on optimizing engineered heparins with improved safety, stability, and bioavailability, and explore the potential of inhaled forms of heparin for antiviral and anti-inflammatory therapies. The development of advanced heparin-based systems, including microneedles, hydrogels, and nanoplatforms for gene delivery, requires overcoming limitations such as thrombogenicity and short half-life. Additionally, addressing concerns related to biosafety, immunogenicity, and stability is critical for improving the clinical translation of heparin-based nanocomposites. Expanding research into alternative heparin sources and synthetic methods is essential to ensure a reliable and cost-effective supply. Integrating these advancements will position heparin at the forefront of innovative therapeutic strategies and biomedical applications.

To balance clinical benefits with potential adverse effects, it is crucial to optimize drug regimens and refine the structure–activity relationships to enhance therapeutic efficacy and safety. Moreover, there is a need for more robust clinical trials to provide reliable evidence for informed medical decision making. Thus, heparin, an esteemed drug with a century-long legacy, has transcended its conventional role as a simple anticoagulant and is poised for further advancement through continued research and optimization efforts.

## Figures and Tables

**Figure 1 pharmaceuticals-17-01362-f001:**
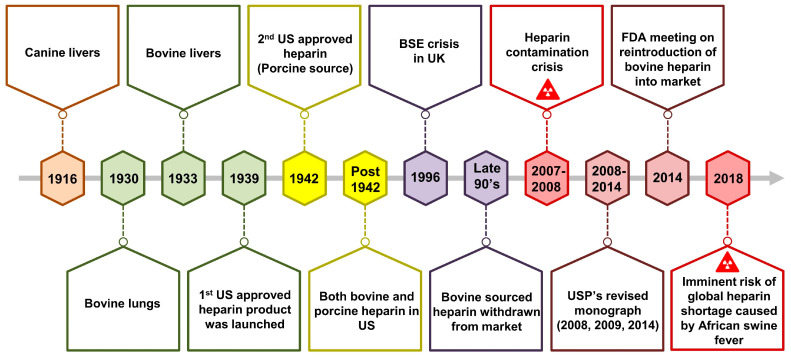
A timeline of the historical development of therapeutic heparin. USP: United States Pharmacopeia; BSE: bovine spongiform encephalopathy; FDA: Food and Drug Administration.

**Figure 2 pharmaceuticals-17-01362-f002:**
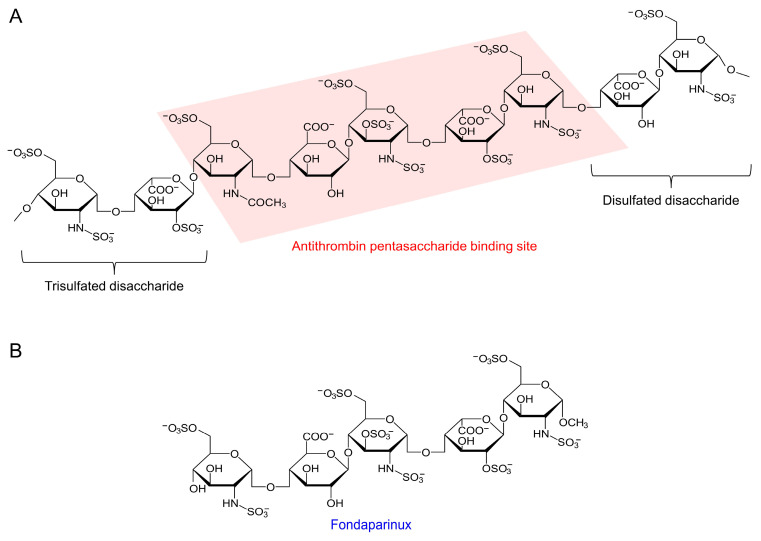
Structural characterization of unfractionated heparin (UFH) and ultra-low molecular weight heparin (ULMWH), Fondaparinux. (**A**) The generalized chemical structure of UFH includes major domains, typically consisting of twenty to fifty copies each of trisulfated and disulfated units. (**B**) Fondaparinux, a synthetic ULMWH, features a specialized antithrombin III (AT) binding site.

**Figure 3 pharmaceuticals-17-01362-f003:**
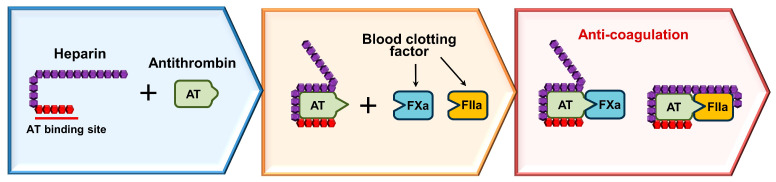
Mechanism of heparin action on blood clotting factors in anti-coagulation. The figure illustrates the structure of the AT-binding pentasaccharide, which is crucial for the inactivation of FXa and factor IIa (FIIa). Longer heparin sequences can further enhance this inactivation.

**Figure 4 pharmaceuticals-17-01362-f004:**
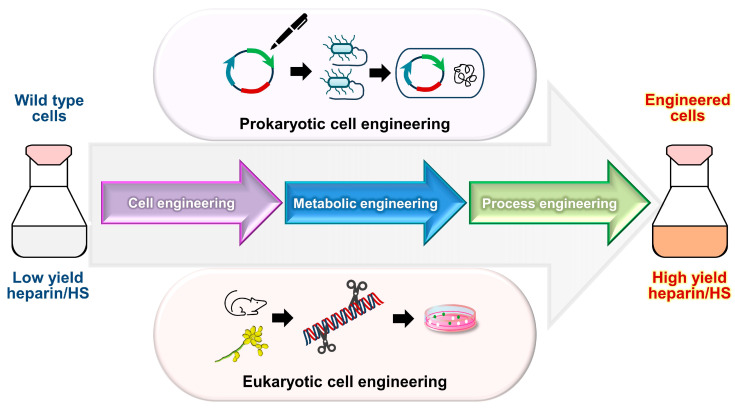
Overview of advanced bioengineering processes in eukaryotic and prokaryotic expression systems, followed by modifications to generate heparin/HS.

**Figure 5 pharmaceuticals-17-01362-f005:**
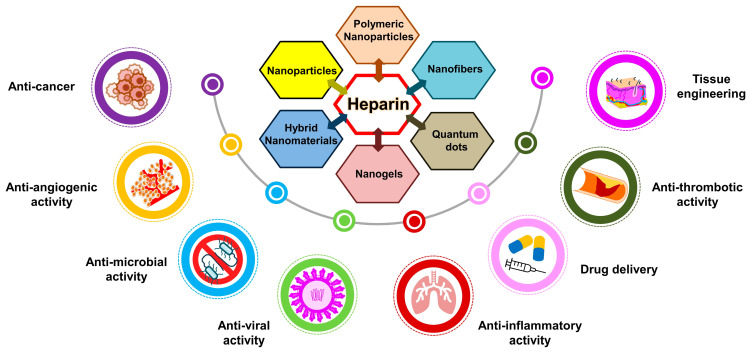
Nanotechnology-driven innovations in heparin-based therapeutics.

**Table 1 pharmaceuticals-17-01362-t001:** Comprehensive overview of LMWHs [2,3,5,50].

LMWHs	Brand Name	Manufacturing	MW (Da)	Anti-Xa:IIa Ratio	Half-Life (hours)	Primary Indications	Major Adverse Effects	Dosing Route	Bioavailability
Tinzaparin	Innohep	Heparinase-induced beta-eliminative cleavage	5500–7500	2.8:1	3–4	VTE and PE	Bleeding and thrombocytopenia	SC	90%
Dalteparin	Fragmin	Deamination-induced cleavage with nitrous acid	5000 (14–26% are >8000)	2.7:1	3–5	DVT, PE, and UA/NSTEMI	Bleeding and thrombocytopenia	SC	87%
Certoparin	Sandoparin	Cleavage through deamination using isoamyl nitrite	5400	2.0–2.2:1	5–6	DVT	Bleeding and thrombocytopenia	SC	>90%
Parnaparin	Fluxum	Copper-catalyzed oxidative depolymerization using hydrogen peroxide	4500	2.3:1	4	DVT, PE, and MI	Bleeding and thrombocytopenia	SC	~100%
Enoxaparin	Lovenox, Clexane	The alkaline-induced cleavage of the benzyl ester of heparin via beta-elimination	2000–8000 (average 4500)	2.7–4:1	4–7	VTE, PE, and ACS	Bleeding and thrombocytopenia	SC	~100%
Reviparin	Clivarin	Deamination-induced cleavage with nitrous acid	4400	4.2:1	3	DVT, PE, and VTE prophylaxis	Bleeding and thrombocytopenia	SC	95%
Nadroparin	Fraxiparin	Deamination-induced cleavage with nitrous acid	5000	3.3:1	3.5	DVT, PE, and VTE prophylaxis	Bleeding and thrombocytopenia	SC	89%
Bemiparin	Beparine	Heparin depolymerized through alkaline degradation	3600	8:1	5–6	DVT, VTE prophylaxis, and ACS	Bleeding and thrombocytopenia	SC	96%
Ardeparin	Normiflo	The peroxide degradation of heparin	5500–6500	1.7–2.4:1	3.3	DVT and PE	Bleeding and thrombocytopenia	SC	92%
**ULMWHs**	**Brand Name**	**Manufacturing**	**MW (Da)**	**Anti-Xa:IIa Ratio**	**Half-Life** **(hours)**	**Primary Indications**	**Major Adverse Effects**	**Dosing Route**	**Bioavailability**
Fondaparinux	Arixtra	Diverse synthetic routes	1508	2–4:1	17–21	DVT, PE, VTE, NSTEMI, STEMI, and UA	Bleeding, thrombocytopenia, and HIT (rare)	SC	~100%
Semuloparin	AVE5026	Selective depolymerization with a phosphazene base induces beta-eliminative cleavage	2000–3000 (average 2400)	80:1	16–20	VTE and PE	Bleeding and thrombocytopenia	SC	90%
RO-14	-	Selective chemical depolymerization induces beta-eliminative cleavage in a non-aqueous medium	1800–3000 (average 2200)	>20:1	8.05	VTE prophylaxis	-	SC	~80–100%

Note: VTE: venous thromboembolism; PE: pulmonary embolism; DVT: deep vein thrombosis; NSTEMI: non-ST-elevation myocardial infarction; STEMI: ST-elevation myocardial infarction; UA: unstable angina; ACS: acute coronary syndrome; SC: subcutaneous.

## Data Availability

Data sharing is not applicable.

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
