# Peer review of "Multifaceted Heparin: Diverse Applications beyond Anticoagulant Therapy"

_pharmaceuticals, 2024, doi:10.3390/ph17101362_

Round 1

Reviewer 1 Report

Comments and Suggestions for Authors

This comprehensive review of applications, innovations, challenges relating to the clinical use of heparin provides a valuable overview of potential developments of an ancient drug. While it is overall well written, a few conceptual issues require further attention. The review should be of major interest to clinicians and pharmacologists.

The various therapeutical effects of heparin are almost exclusively due to interactions, largely electrostatic in nature, with a large variety of proteins that normally bind to HS. The heparin can either substitute for HS, for instance in growth factor action, or displace a protein ligand from immobilized HS-proteoglycan, for instance on an endothelial cell surface. These distinct modes of action should be recognized in the text.

Further, the paragraph between lines 125 and 135 raises structural heterogeneity of heparin as a conceptual issue of importance. The heterogeneity rather applies to HS from different cells and tissues, that are variously designed to accommodate preferential binding to certain protein ligands. Heparin should instead be conceived as a fairly monotonous, almost fully sulfated polymer, capable of interacting with essentially all HS-binding proteins. Heparin thus exposes a variety of binding sequences that are more clearly expressed in cognate HS chains, but masked in heparin by redundant (but tolerated) sulfate groups. Unique binding sequences in heparin require rare substituents, such as the 3-O-sulfated GlcNS residue.

In addition the following issues should be attended to.

In Introduction the work of Robert Rosenberg et al. should be mentioned, in relation to the catalytic role of heparin in activation of antithrombin. Or did I miss that?

Line 94 – The ‘heparan sulfate proteoglycan backbone’ is a misleading expression. Better to directly define the structure of the precursor polymer, N-acetyl heparosan: [4GlcAß1-4GlcNAc1a-]n, also known as the E. coli K5 polysaccharide.

Point out that only about one-third of unfractionated heparin chains contain the antithrombin-binding pentasaccharide sequence.

Lines 141,142 – It is not heparin, but AT that exposes a ‘specialized site’.

Fig. 3 – Shows equally long heparin structures in the inactivation of FXa and FIIa by AT. While it may be true that longer sequences, in addition to the specific AT-binding pentasaccharide sequence may enforce the inactivation of FXa, the presentation fails to illustrate the unique ability of the pentasaccharide alone to catalyze the reaction. Better to show just the pentasaccharide and comment on the catalytic effects of longer chains in the figure legend.

Ll 200,201 – Should better be ‘chemical N-deacetylation and N-sulfation’.

Fig. 4 – The ambition to illustrate bioengineering techniques is laudable but overly simplistic. Can it be improved to provide more tangible information – without getting too complicated?

L 275 – Are refs 77 and 110 adequate in this specific context? The same question applies to L278, refs 26,11,112? A thorough reviewing of all references is recommended.

L 417 – A direct reference to the work of Israel Vlodavsky would be appropriate here.

L 437-449 – Maybe an additional note here to emphasize that DOACs are primarily aimed to prevent thrombosis, but presumably lack other potentially valuable effects of heparin-based drugs of polyanionic nature.

Include a section dealing with the available methods to neutralize overdosed heparin-based anticoagulants – and associated problems?

L 478 – Include the information that sulodexide is actually a mixture of two GAGs, HS and DS.

L 538 – What is bivalirudin?

L 570 – should be HS proteoglycans

In general terms, there is no clear indication of different therapeutic potentials between different LMWH brands. Maybe better to avoid arbitrary emphasis on enoxaparin?

L 651 – Should presumably be ‘heparanase’.

L 656 – Do the authors mean ‘Heparan sulfate (HS)’ rather than ‘Heparin’ here?

L 741, 786 – In spite of improvements, it should be mentioned that the goal, to generate ‘heparin’ by novel methods still has a long way to go before they can replace the drug isolated from animal sources.

L 765 – meaning here of ‘minimal bioavailability’?

Author Response

Comment 1: This comprehensive review of applications, innovations, challenges relating to the clinical use of heparin provides a valuable overview of potential developments of an ancient drug. While it is overall well written, a few conceptual issues require further attention. The review should be of major interest to clinicians and pharmacologists.

Response 1: Thank you for your positive feedback on our review. We appreciated your acknowledgment of its relevance to clinicians and pharmacologists. We carefully addressed the conceptual issues you mentioned to enhance the clarity and depth of the manuscript. Your insights were invaluable in refining our work, and we were committed to making the necessary revisions to meet the expectations of our readership.

Comment 2: The various therapeutical effects of heparin are almost exclusively due to interactions, largely electrostatic in nature, with a large variety of proteins that normally bind to HS. The heparin can either substitute for HS, for instance in growth factor action, or displace a protein ligand from immobilized HS-proteoglycan, for instance on an endothelial cell surface. These distinct modes of action should be recognized in the text.

Response 2: Thank you for your valuable feedback and for highlighting the importance of the electrostatic interactions between heparin and various proteins that normally bind to HS. We agree that these distinct modes of action—where heparin can substitute for HS or displace a protein ligand from immobilized HS-proteoglycan—are critical to understanding heparin’s diverse therapeutic effects. We revised the manuscript to clearly recognize and emphasize these mechanisms in the relevant sections.

Line 127-130 Additionally added paragraph: “The various therapeutic effects of heparin are almost exclusively due to its interactions, largely electrostatic in nature, with a wide variety of proteins that typically bind to HS. Heparin can either substitute for HS, such as in growth factor action, or displace a protein ligand from immobilized HS-proteoglycans, as seen on endothelial cell surfaces.”

Comments 3: Further, the paragraph between lines 125 and 135 raises structural heterogeneity of heparin as a conceptual issue of importance. The heterogeneity rather applies to HS from different cells and tissues, that are variously designed to accommodate preferential binding to certain protein ligands. Heparin should instead be conceived as a fairly monotonous, almost fully sulfated polymer, capable of interacting with essentially all HS-binding proteins. Heparin thus exposes a variety of binding sequences that are more clearly expressed in cognate HS chains, but masked in heparin by redundant (but tolerated) sulfate groups. Unique binding sequences in heparin require rare substituents, such as the 3-O-sulfated GlcNS residue.

Response 3: Thank you for your insightful feedback regarding the structural heterogeneity of heparin and its distinction from HS. We appreciate your clarification on the differences between the structural variability of HS and the relatively monotonous, highly sulfated nature of heparin. Your point about the masking effect of redundant sulfate groups in heparin, and the role of rare substituents in revealing unique binding sequences, is well taken. We have revised the paragraph to better reflect these important distinctions and have incorporated your suggestions to improve the accuracy of our discussion.

Line 131-140 Revised paragraph: “While HS exhibits significant structural heterogeneity across different cells and tissues to accommodate preferential binding to specific protein ligands, heparin is relatively more uniform in structure. It is a highly sulfated polymer capable of interacting with a broad range of HS-binding proteins. The extensive sulfation of heparin exposes a variety of binding sequences that are more distinct and varied in HS but often masked in heparin by redundant sulfate groups. Unique binding sequences in heparin, such as those involving rare substituents like the 3-O-sulfated GlcNS residue, are crucial for its specific biological activities. This structural composition enables heparin to engage in diverse biological interactions, contributing to its broad therapeutic potential beyond anticoagulation, including anti-inflammatory, antiproliferative, and antiviral effects.”

Comment 4: In Introduction the work of Robert Rosenberg et al. should be mentioned, in relation to the catalytic role of heparin in activation of antithrombin. Or did I miss that?

Response 4: Thank you for your suggestion. We will include a reference to Robert Rosenberg et al.'s work on the catalytic role of heparin in antithrombin activation in the Introduction to strengthen the discussion on anticoagulant mechanisms of heparin.

References:

Line 945: 33. Rosenberg, R.; Bauer, K. The heparin-antithrombin system: A natural anticoagulant mechanism. In Hemostasis and Thrombosis: Basic Principles and Clinical Practice; Colman, R.W., Hirsh, J., Marder, V.J., Salzman, E.W., Eds.; J.B. Lippincott & Co.: Philadelphia, PA, U.S.A. 1994, 3, 837–860.

Line 951: 34. Rosenberg, R.D.; Lam, L. Correlation between structure and function of heparin. Proc. Natl. Acad. Sci. U. S. A. 1979, 76, 1218–1222. https://doi.org/10.1073/pnas.76.3.1218.

Comment 5: Line 94 – The ‘heparan sulfate proteoglycan backbone’ is a misleading expression. Better to directly define the structure of the precursor polymer, N-acetyl heparosan: [4GlcAß1-4GlcNAc1a-]n, also known as the E. coli K5 polysaccharide.

Point out that only about one-third of unfractionated heparin chains contain the antithrombin-binding pentasaccharide sequence.

Response 5: Thank you for your insightful comments regarding the terminology used in our manuscript. We have taken your suggestions into account and made the following revisions:

  1. We have changed the expression "heparan sulfate proteoglycan backbone" to more accurately define the structure of the precursor polymer as N-acetyl heparosan, represented as [4GlcA β1,4GlcNAc α1,]n, also known as the Escherichia coli K5 polysaccharide.
  2. We have added clarification that only about one-third of unfractionated heparin chains contain the antithrombin-binding pentasaccharide sequence, highlighting the importance of this sequence in heparin's anticoagulant activity.

Line: 93-98 Revised paragraph: “Heparin and heparan sulfate (HS) are structurally similar glycosaminoglycans (GAGs) synthesized from a shared precursor polymer, N-acetyl heparosan, with the repeating disaccharide structure [4GlcA β1,4GlcNAc α1,]n, also known as the Escherichia coli (E. coli) K5 polysaccharide. Only about one-third of unfractionated heparin chains contain the AT-binding pentasaccharide sequence, which is critical for its anticoagulant activity.”

Comment 6: Lines 141,142 – It is not heparin, but AT that exposes a ‘specialized site’.

Response 6: Thank you for your valuable comment regarding the binding mechanism of heparin and AT. We appreciate your clarification and have revised the sentence accordingly to accurately reflect that it is AT that exposes the specialized site.

Line 146-147 Revised sentence: “Upon binding to AT, heparin facilitates the exposure of a specialized site on AT that selectively targets and inhibits FXa.”

Comment 7: Fig. 3 – Shows equally long heparin structures in the inactivation of FXa and FIIa by AT. While it may be true that longer sequences, in addition to the specific AT-binding pentasaccharide sequence may enforce the inactivation of FXa, the presentation fails to illustrate the unique ability of the pentasaccharide alone to catalyze the reaction. Better to show just the pentasaccharide and comment on the catalytic effects of longer chains in the figure legend.

Response 7: Thank you for your insightful comment regarding Figure 3. We appreciate your suggestion to clarify the representation of heparin structures in the context of FXa and FIIa inactivation by AT.

In response to your feedback, we revised Figure 3 to display only the antithrombin-binding pentasaccharide. Additionally, we updated the figure legend to include a discussion of how longer heparin sequences can enhance the inactivation of FXa while emphasizing the unique catalytic ability of the pentasaccharide alone.

Comment 8: Ll 200,201 – Should better be ‘chemical N-deacetylation and N-sulfation’.

Response 8: Thank you for your helpful suggestion. We will revise the text to accurately state "chemical N-deacetylation and N-sulfation" as you recommended.

Line 205-206 Revised line: “Initially, the polysaccharide, also known as heparosan, undergoes chemical N-deacetylation and N-sulfation.”

Comment 9: Fig. 4 – The ambition to illustrate bioengineering techniques is laudable but overly simplistic. Can it be improved to provide more tangible information – without getting too complicated?

Response 9: Thank you for your valuable feedback on Figure 4. We acknowledged that the original illustration was overly simplistic and have revised it to include more tangible information regarding bioengineering techniques while maintaining clarity without unnecessary complexity.

Comment 10: L 275 – Are refs 77 and 110 adequate in this specific context? The same question applies to L278, refs 26,11,112? A thorough reviewing of all references is recommended.

Response 10: Thank you for your comment. We have thoroughly reviewed references 77, 110, 26, 11, and 112 for their adequacy in context. We ensured that all cited references are relevant and appropriate, making necessary adjustments to enhance the manuscript's overall quality.

Added references:

  1. Linhardt, R.J.; Liu, J. Synthetic heparin. Curr. Opin. Pharmacol., 2012, 12, 217-9, https://doi.org/10.1016/j.coph.2011.12.002.
  2. Ding, Y.; Prasad, C.V.V.; Bai, H.; Wang, B. Efficient and practical synthesis of Fondaparinux. Bioorg. Med. Chem. Lett. 2017, 27, 2424-2427, https://doi.org/10.1016/j.bmcl.2017.04.013.
  3. Dey, S.; Lo, H.J.; Wong, C.H. Programmable one-pot synthesis of heparin pentasaccharide fondaparinux. Org. lett. 2020, 22, 4638-4642, https://doi.org/10.1021/acs.orglett.0c01386.
  4. Jin, H.; Chen, Q.; Zhang, Y.Y.; Hao, K.F.; Zhang, G.Q.; Zhao, W. Preactivation-based, iterative one-pot synthesis of anticoagulant pentasaccharide fondaparinux sodium. Org. Chem. Front.2019, 6, 3116-3120, https://doi.org/10.1039/C9QO00480G.

Comment 11: L 417 – A direct reference to the work of Israel Vlodavsky would be appropriate here.

Response 11: Thank you for your suggestion. We have added a direct reference to the work of Israel Vlodavsky in the text at line 417 to enhance the discussion and acknowledge his contributions.

Added reference:

  1. Vlodavsky, I.; Abboud-Jarrous, G.; Elkin, M.; Naggi, A.; Casu, B.; Sasisekharan, R.; Ilan, N. The impact of heparanese and heparin on cancer metastasis and angiogenesis. Pathophysiol. Haemost. Thromb.2006, 35, 116–127. https://doi.org/10.1159/000093553.

Comment 12: L 437-449 – Maybe an additional note here to emphasize that DOACs are primarily aimed to prevent thrombosis, but presumably lack other potentially valuable effects of heparin-based drugs of polyanionic nature.

Response 12: Thank you for your insightful comment. We have added a note to emphasize that while DOACs are primarily aimed at preventing thrombosis, they likely lack other potentially valuable effects associated with heparin-based drugs of polyanionic nature.

Line 451-452: “While DOACs are primarily aimed at preventing thrombosis, they presumably lack other potentially valuable effects of heparin-based drugs of polyanionic nature.”

Comment 13: Include a section dealing with the available methods to neutralize overdosed heparin-based anticoagulants – and associated problems?

Response 13: Thank you for your insightful comment. We included a section in the manuscript that discusses the available methods for neutralizing overdosed heparin-based anticoagulants, along with the potential problems associated with these methods. This addition enhanced the comprehensiveness of the manuscript regarding the management of heparin therapy.

 Line 458-471 Added paragraph: “Several methods are available to address the neutralization of overdosed heparin-based anticoagulants, each with its own limitations. Protamine sulfate is the primary agent used to neutralize UFH by forming a stable complex with heparin, effectively reversing its anticoagulant effects. However, when used for LMWHs, protamine only partially neutralizes their activity, typically reversing about 60-80%, leaving residual anticoagulant effects that can pose a clinical challenge, especially in cases of significant overdose. Additionally, protamine sulfate can cause adverse reactions, such as hypotension, bradycardia, and anaphylaxis, particularly in patients with fish allergies or previous exposure to protamine. Alternative approaches, such as hemodialysis or the use of agents like recombinant factor VIIa or activated prothrombin complex concentrates, have been explored for reversing heparin effects, but they are less effective and carry their own risks. These limitations underscore the complexity of managing heparin overdoses and highlight the need for careful monitoring and individualized treatment plans to mitigate the risks associated with both heparin use and its neutralization [196].”

  1. Frackiewicz, A.; Kalaska, B.; Miklosz, J.; Mogielnicki, A. The methods for removal of direct oral anticoagulants and heparins to improve the monitoring of hemostasis: a narrative literature review. Thromb. J.2023, 21, 58. https://doi.org/10.1186/s12959-023-00501-7.

 Comment 14: L 478 – Include the information that sulodexide is actually a mixture of two GAGs, HS and DS.

Response 14: Thank you for your suggestion. We included the information that sulodexide is a mixture of two GAGs, HS and dermatan sulfate (DS), in the revised manuscript. This clarification enhanced the accuracy of the content.

Line 499-503: “Recent clinical investigations highlight the promising role of sulodexide, a LMW glycosaminoglycan that is actually a mixture of two GAGs —HS and dermatan sulfate (DS) — in ameliorating proteinuria in DN patients, even when co-administered with angiotensin-converting enzyme (ACE) inhibitors or angiotensin receptor antagonists, suggesting its potential for renal protection.”

Comment 15: L 538 – What is bivalirudin?

Response 15: Line 561-563: “Bivalirudin is a synthetic direct thrombin inhibitor used as an anticoagulant, primarily in patients undergoing percutaneous coronary intervention (PCI) for acute coronary syndromes, particularly ST-elevation myocardial infarction (STEMI).”

Comment 16: L 570 – should be HS proteoglycans

Response 16: Thank you for your suggestion. The term has been revised to "HS proteoglycans" at line 570 to ensure accuracy and clarity in the manuscript.

Line 593-595: “Recent research has elucidated the role of HS proteoglycans in AD pathogenesis [222], influencing the formation of plaques and tangles while facilitating pathogen entry into brain cells.”

Comment 17: In general terms, there is no clear indication of different therapeutic potentials between different LMWH brands. Maybe better to avoid arbitrary emphasis on enoxaparin?

Response 17: Thank you for your valuable feedback. We have revised the paragraph to present a more balanced view regarding the therapeutic potentials of different low molecular weight heparins (LMWHs) and minimized the specific emphasis on enoxaparin.

Line 599-600: “Notably, certain LMWHs have shown promise in reducing beta-amyloid plaque accumulation and enhancing cognitive function in Alzheimer's disease (AD) animal models.”

Comment 18: L 651 – Should presumably be ‘heparanase’.

Response 18: We appreciate the reviewer’s comment and have corrected "heparinase" to "heparanase" in the revised text.

Revised line 673: “In the realm of metastasis, tumor cells exhibiting heightened heparanase expression facilitate epithelial-mesenchymal transition (EMT), a pivotal process for hematogenous metastasis.”

Comment 19: L 656 – Do the authors mean ‘Heparan sulfate (HS)’ rather than ‘Heparin’ here?

Response 19: We appreciate the reviewer’s observation. We have revised the sentence to clarify that it refers to "Heparan sulfate (HS)" rather than "Heparin." The updated sentence now reads:

Line 678 “HS emerges as a key contributor in angiogenesis, navigating through a complicated path affected by chain length and sulfation position.”

Comment 20: L 741, 786 – In spite of improvements, it should be mentioned that the goal, to generate ‘heparin’ by novel methods still has a long way to go before they can replace the drug isolated from animal sources.

Response 20: We appreciate the reviewer’s insightful comment regarding the challenges in generating heparin through novel methods. Below are the revised sentences that incorporate this perspective:

Line 763-767: Revised Sentence 1: “The complex extraction process of crude heparin from animal sources remains a limitation compared to other biopolymers. Although significant strides have been made in depolymerization, chemical modification, and synthetic production, it is important to recognize that achieving a sustainable alternative to animal-derived heparin still has a long way to go.”

Line 810-814: Revised Sentence 2: “These engineered heparins have the potential to fine-tune heparin-binding motifs and other molecular characteristics, thereby enhancing therapeutic efficacy and reducing side effects. However, the development of these alternatives is still ongoing and requires further research to fully realize their potential.”

Comment 21: L 765 – meaning here of ‘minimal bioavailability’?

Response 21: Thank you for your insightful comment. We clarified the meaning of “minimal bioavailability” in the revised sentence, specifying that it refers to the limited amount of the drug that reaches systemic circulation after administration.

Revised line 787-790: “Furthermore, the low bioavailability of heparin and its low molecular weight counterparts, which refers to the limited amount of the drug reaching systemic circulation after administration, presents a significant challenge that can be effectively addressed by incorporating them into nanocomposites.”

Reviewer 2 Report

Comments and Suggestions for Authors

The review submitted by Sultana and Kanihira describes applications of heparin derivatives beyond their anticoagulant activity. First, it describes the structure of heparin, its different types and forms, the bases of its anticoagulant activity, and its different applications. Subsequently, it focuses on obtaining the natural product through different alternatives to the extraction of the natural product: Chemical Synthesis, Chemoenzymatics, and advances in bioengineering. The introduction ends with a section describing the primary indications and main adverse effects of LMWHs preparations. According to the title until here is the introduction and now begins the non-anticoagulant applications

This section begins with the description of anti-inflammatory applications, which are listed and examples in which anti-inflammatory effects not associated with antithrombotics have been described using derivatives known as antithrombotic affinity-depleted heparin that cannot associate with AT-III.

Next is the activity in COVID-19 and other viral infections. First, it describes thrombosis prevention after the antiviral activity is found. 

Continue with use in COVID-19 and other infectious diseases. Describes the benefit as antithrombotic therapy and the reduction of inflammation associated with COVID-19 infection by interacting with cytokines. Then they described the use of heparin as a blocker of infections by various viruses, such as Denge, Herpes, HIV, and Zika. Another disease that can be treated with heparin is malaria.

In many other diseases, oncology, nephropathy, cardiopathy and neuroprotection, positive effects of heparin are also described.

Next, heparin's applications in nanomedicine are described as stabilizers and biocompatibilizers of nanoparticles for medicine and Smart drug delivery systems.

Finally, it presents a series of improvements that will guide future research on improving the use of heparin in the applications described.

Author Response

Comments: The review submitted by Sultana and Kanihira describes applications of heparin derivatives beyond their anticoagulant activity. First, it describes the structure of heparin, its different types and forms, the bases of its anticoagulant activity, and its different applications. Subsequently, it focuses on obtaining the natural product through different alternatives to the extraction of the natural product: Chemical Synthesis, Chemoenzymatics, and advances in bioengineering. The introduction ends with a section describing the primary indications and main adverse effects of LMWHs preparations. According to the title until here is the introduction and now begins the non-anticoagulant applications.

This section begins with the description of anti-inflammatory applications, which are listed and examples in which anti-inflammatory effects not associated with antithrombotics have been described using derivatives known as antithrombotic affinity-depleted heparin that cannot associate with AT-III.

Next is the activity in COVID-19 and other viral infections. First, it describes thrombosis prevention after the antiviral activity is found. 

Continue with use in COVID-19 and other infectious diseases. Describes the benefit as antithrombotic therapy and the reduction of inflammation associated with COVID-19 infection by interacting with cytokines. Then they described the use of heparin as a blocker of infections by various viruses, such as Denge, Herpes, HIV, and Zika. Another disease that can be treated with heparin is malaria.

In many other diseases, oncology, nephropathy, cardiopathy and neuroprotection, positive effects of heparin are also described.

Next, heparin's applications in nanomedicine are described as stabilizers and biocompatibilizers of nanoparticles for medicine and Smart drug delivery systems.

Finally, it presents a series of improvements that will guide future research on improving the use of heparin in the applications described.

Response: Thank you for your thorough and insightful review of our manuscript. We greatly appreciate your detailed analysis and recognition of the key aspects of our work, including the structure of heparin, its anticoagulant activity, and its wide range of non-anticoagulant applications.

Once again, thank you for your valuable feedback and for taking the time to review our manuscript. We are confident that your comments will further enhance the contribution of this review to the glycobiology and biomedical research communities.

Reviewer 3 Report

Comments and Suggestions for Authors

This is an excellent review paper on the diverse biomedical applications of heparin. It presents an interesting topic, and the information is valuable for glycobiology and heparin-related biomedical research. The review is well-organized and well-written. This reviewer recommends accepting this manuscript as it is.

Author Response

Comments: This is an excellent review paper on the diverse biomedical applications of heparin. It presents an interesting topic, and the information is valuable for glycobiology and heparin-related biomedical research. The review is well-organized and well-written. This reviewer recommends accepting this manuscript as it is.

Response: Thank you very much for your positive and encouraging feedback on our manuscript. We are delighted that you found the topic and content valuable for glycobiology and heparin-related biomedical research. Your recognition of our work's organization and writing is deeply appreciated. We are grateful for your recommendation to accept the manuscript, and we are confident that your feedback will further motivate our future research in this field. Thank you once again for your thoughtful review.